# Antitumour Effects of Astaxanthin and Adonixanthin on Glioblastoma

**DOI:** 10.3390/md18090474

**Published:** 2020-09-18

**Authors:** Shohei Tsuji, Shinsuke Nakamura, Takashi Maoka, Tetsuya Yamada, Takahiko Imai, Takuya Ohba, Tomohiro Yako, Masahiro Hayashi, Ken Endo, Masanao Saio, Hideaki Hara, Masamitsu Shimazawa

**Affiliations:** 1Department of Biofunctional Evaluation, Molecular Pharmacology, Gifu Pharmaceutical University, Gifu 501-1196, Japan; tsuji.yakkou@gmail.com (S.T.); yamada.yakkou@gmail.com (T.Y.); imai.yakkou@gmail.com (T.I.); ohba.yakkou@gmail.com (T.O.); yako.yakkou@gmail.com (T.Y.); hidehara@gifu-pu.ac.jp (H.H.); shimazawa@gifu-pu.ac.jp (M.S.); 2Research Institute for Production Developent Division of Food Function and Chemistry, Kyoto 606-0805, Japan; maoka@mbox.kyoto-inet.or.jp; 3Department of Neurosurgery, Gifu University School of Medicine, Gifu 501-1194, Japan; 4Department of HPM Research & Development, Biotechnology R&D Group, High Performance Materials Company, ENEOS Corporation, Yokohama 231-0815, Japan; hayashi.masahiro@eneos.com; 5Department of HPM Business Promotion Group V, Business promotion Group, High Performance Materials Company, ENEOS Corporation, Tokyo 108-8005, Japan; endo.ken@eneos.com; 6Graduate School of Health Sciences, Gunma University, Gunma 371-8514, Japan; saio@gunma-u.ac.jp

**Keywords:** brain, cancer, oral administration, *paracoccsu carotinifaciens*, xanthophyll carotenoid

## Abstract

Several antitumour drugs have been isolated from natural products and many clinical trials are underway to evaluate their potential. There have been numerous reports about the antitumour effects of astaxanthin against several tumours but no studies into its effects against glioblastoma. Astaxanthin is a red pigment found in crustaceans and fish and is also synthesized in *Haematococcus pluvialis*; adonixanthin is an intermediate product of astaxanthin. It is known that both astaxanthin and adonixanthin possess radical scavenging activity and can confer a protective effect on several damages. In this study, we clarified the antitumour effects of astaxanthin and adonixanthin using glioblastoma models. Specifically, astaxanthin and adonixanthin showed an ability to suppress cell proliferation and migration in three types of glioblastoma cells. Furthermore, these compounds were confirmed to transfer to the brain in a murine model. In the murine orthotopic glioblastoma model, glioblastoma progression was suppressed by the oral administration of astaxanthin and adonixanthin at 10 and 30 mg/kg, respectively, for 10 days. These results suggest that both astaxanthin and adonixanthin have potential as treatments for glioblastoma.

## 1. Introduction

Glioblastoma is one of the most lethal types of brain tumour; it arises from glial cells [1,2]. The standard treatment for glioblastoma is a combination of chemotherapy and radiotherapy following the surgical removal of tumour tissue [3]. However, glioblastomas generally show a poor prognosis and short survival time, rarely longer than 14 months [4]. The poor prognosis is attributed to chemoresistance to temozolomide, the first-line drug for the treatment of glioblastoma [5]. Therefore, it is essential that novel drugs are developed that possess an antitumour effect based on different mechanisms to those of temozolomide. However, the development of novel drugs for glioblastoma has been limited by the blood–brain barrier (BBB) issue [6]. Recent studies have shown that certain compounds derived from natural products, such as curcumin and resveratrol, can cross the BBB and have an antitumour effect against glioblastoma [7,8]. Both powerful antitumour effects and transferability to brain tissues are essential requirements for any new treatments for glioblastoma.

In this study, we focused on a xanthophyll carotenoid, astaxanthin, and its intermediate product, adonixanthin (Figure 1) [9]. Astaxanthin, a red pigment that occurs naturally in shrimp, crab, and salmon [10], is a powerful antioxidant and has shown some protective effects in various oxidative stress and disease models [11,12,13,14,15]. Adonixanthin has similarly powerful antioxidative effects [9,16]. Crucially, it has been reported that, in mouse models, both astaxanthin and adonixanthin have the ability to cross the BBB to reach the brain tissue and can protect vessels in the brain from cerebral ischaemia and haemorrhage [17,18]. However, there have been no reports that either astaxanthin or adonixanthin have antitumour effects against glioblastoma. It is known that astaxanthin has antitumour effects against oral cancer, bladder carcinogenesis, colon carcinogenesis, leukaemia, and hepatocellular carcinoma [19,20,21,22,23]; however, the mechanisms for this antitumour activity of astaxanthin are yet to be fully clarified. It has been reported that the antitumour effect of astaxanthin and adonixanthin are mediated by multiple mechanisms, including JAK-2/STAT-3, NF-κB, ERK, AKT (PKB), PPARγ, and Nrf2 [24].

The purpose of this study was to clarify whether astaxanthin and adonixanthin have antitumour effects against glioblastoma following their oral administration. Furthermore, we aimed to verify whether orally administered astaxanthin or adonixanthin can be absorbed by the brain tissue. We investigated the antitumour mechanisms of astaxanthin and adonixanthin using glioblastoma cells.

## 2. Results

### 2.1. Astaxanthin and Adonixanthin Suppressed the Growth of Glioblastoma Cells

Robust cell viability is an important characteristic of tumour cells. We performed a cell viability assay using the murine glioblastoma cell line GL261 and the human glioblastoma cell line U251MG. Both astaxanthin and adonixanthin showed antitumour effects against GL261 and U251MG cells, in a concentration-dependent manner (Figure 2A,B). In GL261 cells, astaxanthin and adonixanthin suppressed cell viability at concentrations of more than 5 and 0.1 µM (Figure 2A). In U251MG cells, astaxanthin and adonixanthin suppressed cell viability at concentrations of more than 1 and 0.1 µM (Figure 2B). Moreover, we performed the BrdU cell proliferation assay to determine whether the results of cell viability by astaxanthin and adonixanthin (Figure 2A,B) are based on proliferation. Astaxanthin and adonixanthin at the 10 µM treatment for 72 h reduced the number of BrdU-positive cells (Figure 2C,D). The BrdU-positive cell rates were 60.56 ± 1.40% (control group), 19.17 ± 1.77% (temozolomide group), 49.13 ± 2.66% (astaxanthin group), and 41.31 ± 1.51% (adonixanthin group).

### 2.2. Astaxanthin and Adonixanthin Suppressed the Migration of Glioblastoma Cells

In addition to cell viability, cell migration is also important for tumour enlargement. We performed a wound healing assay using the murine glioblastoma cell line GL261 and the human glioblastoma cell line U251MG. Astaxanthin and adonixanthin suppressed cell migration by 8.84 ± 2.09% and 13.74 ± 4.01% in GL261, respectively (Figure 3A,B). Astaxanthin and adonixanthin suppressed cell migration by 21.15 ± 1.84% and 26.32 ± 5.14% in U251MG, respectively, at a concentration of 10 µM (Figure 3C,D).

### 2.3. Astaxanthin and Adonixanthin Decreased the Expression of Some Proteins to Promote Cell Growth and Migration

We performed immunoblotting to reveal the mechanisms of the antitumour effects of astaxanthin and adonixanthin. In general, the phosphorylation of ERK1/2 and Akt are accelerated during tumour progression in many types of tumour [25,26]. The phosphorylation of ERK1/2 and Akt were decreased 6 h post-treatment with astaxanthin and adonixanthin (Figure 4A,B). Both astaxanthin and adonixanthin treatment for 48 h increased the phosphorylation of p38 mitogen-activated protein kinase: MAPK (Figure 4C). To elucidate the antitumour effect of astaxanthin and adonixanthin, we confirmed the expression of cyclin D1 (cell cycle-related protein) and p27 (cyclin-dependent kinase inhibitor). As a result, both astaxanthin and adonixanthin treatment for 48 h decreased the expression of cyclinD1 (Figure 4D) and increased the expression of p27 (Figure 4E). Next, we focused on some proteins that are related to cell migration (Matrix metalloproteinase-2, -9: MMP-2, -9, and fibronectin). Treatment with astaxanthin and adonixanthin for 48 h decreased MMP-2 expression (Figure 4F) but not MMP-9 expression (Figure 4G). Interestingly, adonixanthin also decreased the expression of fibronectin (Figure 4H).

### 2.4. Astaxanthin and Adonixanthin Decreased the Production of Reactive Oxygen Species (ROS)

In previous reports, the presence of ROS was shown to promote tumour progression via the phosphorylation of ERK1/2 and Akt [27,28]. Therefore, we performed an ROS assay to investigate the effects of astaxanthin and adonixanthin on the level of intracellular ROS. Both astaxanthin and adonixanthin decreased the amount of intracellular ROS, with the effect of adonixanthin stronger than that of astaxanthin (Figure 5A). We also verified the expression of NADPH oxidase 4 (Nox4), which is a representative factor to produce ROS. Adonixanthin significantly decreased the expression of Nox4, whereas astaxanthin produced no change in Nox4 expression (Figure 5B).

### 2.5. Concentrations of Astaxanthin and Adonixanthin in Murine Serum and Tissues

Following the oral administration of astaxanthin, the concentration of *trans*- or *cis*-astaxanthin in the serum was 6.04 ± 1.16 and 13.35 ± 3.78 ng/mL, respectively. Astaxanthin was not detected in the serum in pretreatment samples (Table 1). Adonixanthin was also only detected in the treated group. The levels of *trans*- and *cis*- adonixanthin were 34.72 ± 3.00 and 35.02 ± 5.01 ng/mL, respectively (Table 1). In the adonixanthin-treated group, the *trans*-isomer was detected significantly more frequently compared with its detection in the astaxanthin-treated group.

As shown in Table 2, following oral administration of astaxanthin or adonixanthin, these compounds were absorbed in each region of the brain in mice. *Trans*-astaxanthin was detected at 5.22 ± 0.87, 5.06 ± 1.85, 11.37 ± 3.14, and 12.28 ± 1.07 ng/g in the cerebral cortex, the cerebellum, the striatum, and the hippocampus of astaxanthin-treated mice, respectively. *Cis*-astaxanthin was detected at 3.62 ± 1.56, 2.43 ± 1.58, 3.76 ± 2.06, and 2.61 ± 2.67 ng/g in the cerebral cortex, the cerebellum, the striatum, and the hippocampus of astaxanthin-treated mice, respectively. *Trans*-adonixanthin was detected at 3.24 ± 0.44, 2.85 ± 1.05, 2.90 ± 1.87, and 3.63 ± 2.10 ng/g in the cerebral cortex, the cerebellum, the striatum, and the hippocampus of adonixanthin-treated mice, respectively. There was no *cis*-astaxanthin in any of the brain tissues. Furthermore, neither astaxanthin nor adonixanthin were detected in the brains of the vehicle-treated group.

### 2.6. The Effects of Astaxanthin and Adonixanthin on Body Weight

To investigate systemic influences following the administration of astaxanthin and adonixanthin, we explored changes in the body weight of mice. The weight of vehicle-, astaxanthin-, and adonixanthin-treated mice at pretreatment was 34.33 ± 2.04, 34.28 ± 1.93, and 34.22 ± 2.67 g, respectively. After 10 days of administration, the weight of vehicle-, astaxanthin-, and adonixanthin-treated mice was 34.72 ± 2.05, 35.50 ± 1.95, and 36.15 ± 2.05 g, respectively (Appendix A). No significant changes in body weight were observed in mice treated with either astaxanthin or adonixanthin.

### 2.7. Astaxanthin and Adonixanthin Showed Antitumour Effects in a Murine Orthotopic Glioblastoma Model

We investigated whether astaxanthin and adonixanthin exhibited antitumour effects in a murine orthotopic glioblastoma model. Treatment with astaxanthin and adonixanthin at 10 and 30 mg/kg for 10 days, respectively, significantly suppressed tumour enlargement (Figure 6A,B).

## 3. Discussion

Astaxanthin and adonixanthin, synthesized in marine organisms [10], have antitumour properties [9] and a therapeutic effect on the central nervous system [18,29]. However, there are no reports about their effects for glioma. In the present study, we demonstrated the antitumour effects of these compounds in both in vitro and in vivo glioblastoma models.

Astaxanthin and adonixanthin inhibited both cell proliferation and migration in human and mouse glioblastoma cells (Figure 2 and Figure 3). Next, in order to elucidate the antitumour mechanism of astaxanthin and adonixanthin, the expression of proteins related to tumour progression and the degree of ROS production were examined using the mouse glioblastoma cell line GL261. Astaxanthin and adonixanthin were found to reduce the expression of phosphorylated ERK1/2 and phosphorylated Akt (Figure 3A,B). It was shown that astaxanthin exhibited an antitumour effect in an oral cancer model via the suppression of phosphorylation of ERK1/2 and Akt [19]. Similarly, it is presumed that astaxanthin and adonixanthin also have an antitumour effect against glioblastoma through the inhibition of the phosphorylation of ERK1/2 and Akt. Furthermore, astaxanthin and adonixanthin increased the expression of phosphorylated p38 (Figure 4C). The phosphorylation of p38 can lead to cell damage and cell cycle arrest [30]. Therefore, the antitumour effects of both astaxanthin and adonixanthin were involved in increasing the expression of phosphorylated p38. To elucidate the antitumour effect of these compounds, we confirmed the expression of cell cycle-related protein cyclin D1, and apoptosis-related protein Bcl-2. These compounds decreased the expression of cyclin D1 (Figure 4D) but not Bcl2 (Appendix A). In fact, these compounds did not induce cell death in GL261 (Appendix A) and decreased the cell proliferation in GL261 (Figure 2C,D). Moreover, these compounds increased the expression of p27, a cyclin-dependent kinase inhibitor (Figure 4E). In a previous report, the phosphorylation of ERK1/2 and Akt increased the expression of cyclinD1 and decreased the expression of p27 [31,32,33]. These results indicate that the antitumour effect of both astaxanthin and adonixanthin may be mediated by not cell death but cell cycle arrest. Temozolomide significantly reduced the expression of p27 (Figure 4E). This may be due to a feedback to the potent cell cycle arrest effect of temozolomide, as previously reported [34]. Additionally, adonixanthin reduced the expression of MMP-2 and fibronectin, downstream of ERK1/2 and Akt signalling (Figure 4E,G). In addition, both astaxanthin and adonixanthin decreased the mRNA level of fibronectin (Appendix A). These results indicate that adonixanthin could affect ERK1/2 and Akt signalling upstream. In MMP9, the reason why there were no changes in the expression (Figure 3F) may be that it is an inflammation-related enzyme, the expression of which is low in the normal condition.

To elucidate the active site of both compounds, we examined their effect on ROS, which is important for the regulation of both ERK1/2 and Akt phosphorylation [35]. In the past, it has been reported that ROS promote tumour progression via the phosphorylation of ERK1/2 and Akt [27,30]. Therefore, we evaluated the level of ROS in glioblastoma cells following treatment with both compounds for 6 h. Both compounds greatly reduced the levels of intracellular ROS (Figure 5A). Astaxanthin and adonixanthin have been reported to possess radical scavenging properties [9], and the reduction of reactive oxygen species in glioblastoma cells shown in this study may also include the direct antioxidant activity of these compounds. Nox4 is a key factor involved in the regulation of ROS production and is upregulated in glioblastoma compared with other nicotinamide adenine dinucleotide phosphate: NADPH oxidase isoforms [36]. Only adonixanthin significantly suppressed the expression of Nox4 (Figure 5B). As the effect of adonixanthin on intracellur ROS in glioblastoma cells, it is considered that adonixanthin may inhibit the expression of ROS production-related factors, such as Nox4.

We examined whether astaxanthin and adonixanthin can be delivered to the brain following oral administration, using healthy mice. We confirmed that both compounds were delivered to the brain, that astaxanthin was detected at a higher concentration than adonixanthin in the brain tissue, and that the *cis-*form of adonixanthin was not detected at all in any brain tissues (Table 2). Conversely, adonixanthin was detected at high levels in mouse tissues other than the brain (Appendix A). It is suggested that orally administered adonixanthin mainly affects the peripheral tissues due to the difference in the distribution of adonixanthin. In this study, we used structurally stable *trans*-isomers of both astaxanthin and adonixanthin. The *trans*-form of astaxanthin and adonixanthin is converted to the *cis*-form in the blood following oral administration [37]. The differences in the structure between the *cis*- and *trans*-forms may affect their antitumour activity, and although the *cis*-form of astaxanthin has been reported to show greater antioxidant activity than the *trans*-form [38], the detailed mechanism underlying this remains unknown. In addition, the concentrations of astaxanthin and adonixanthin detected in the brain were approximately 30 and 10 nM, respectively. These concentrations correspond to one-third and one-tenth of the minimum concentration (0.1 μM) used in the cell proliferation test, as shown in Figure 2A,B. In glioblastoma pathology, invasive glioblastoma cells degrade the basement membrane around blood vessels and cause disruption of the blood–brain barrier. As a result, the transferability to the brain of immune cells and chemotherapeutic drugs is increased [39,40]. In a study using a glioma rat model, it was reported that translocation of a magnetic resonance imaging (MRI) contrast agent increased about five times in tumour tissues compared with its translocation in healthy tissues [41]. Therefore, it is inferred that astaxanthin and adonixanthin administered orally accumulate in glioblastoma tissues at higher concentrations compared with their accumulation in healthy tissues.

Next, using an in vivo glioblastoma mouse model, we examined whether the oral administration of astaxanthin and adonixanthin exhibited an antitumour effect on glioblastoma. Both astaxanthin and adonixanthin significantly suppressed tumour growth in this in vivo glioblastoma model (Figure 6). These results showed that astaxanthin and adonixanthin transferred to the brain by oral administration exert an antitumour effect on glioblastoma. Although adonixanthin tended to have a greater effect than astaxanthin in this in vitro study (Figure 2, Figure 3, Figure 4 and Figure 5), astaxanthin and adonixanthin showed a comparable antitumour effect in the in vivo glioblastoma model. These results may reflect differences both in transferability to the brain and the ratio of isoforms of astaxanthin or adonixanthin.

In conclusion, these findings suggest that the oral administration of astaxanthin and adonixanthin, respectively, could be potentially useful treatments for glioblastoma.

## 4. Materials and Methods

### 4.1. Reagents

Both astaxanthin and adonixanthin obtained from *Paracoccsu carotinifaciens* were provided by ENEOS Corporation (Tokyo, Japan). Adonixanthin is an intermediate compound between zeaxanthin and astaxanthin [42,43]. These compounds were dissolved by dimethyl sulfoxide, DMSO (FUJIFILM Wako Pure Chemical Corporation, Osaka, Japan), on in vitro and diffused by olive oil (FUJIFILM Wako Pure Chemical Corporation, Osaka, Japan) on in vivo. In the in vitro study, the final concentration of DMSO in all the groups was 0.1%.

### 4.2. Cell Line and Culture Condition

The human glioblastoma cell line U251MG was purchased from European Collection of Authenticated Cell Cultures (ECACC; London, the United Kingdom). The murine glioblastoma cell line GL261 was kindly provided by Dr. Saio, Graduate School of Health Sciences, Gunma University. The human glioblastoma cell line U87MG was obtained from American Type Culture Collection (ATCC). These cells were cultured in Dulbecco’s Modified Eagle Medium (DMEM) with low glucose (Nacalai Tesque, Tokyo, Japan) supplemented with 10% foetal bovine serum (FBS; Valeant, Costa Mesa, CA, USA), 100 units/mL penicillin, and 100 mg/mL streptomycin at 37 °C in 5% CO₂. Cells were passaged by trypsinization and used within 10 passages.

### 4.3. Cell Viability

U251MG, GL261, or U87MG cells were seeded onto 96-well plates at a density of 2 × 10^3^ cells/well with DMEM supplemented with 10% FBS and then incubated for 24 h, after which the culture medium was changed to DMEM containing 10% FBS. Then, astaxanthin, adonixanthin or temozolomide (Tokyo Chemical Industry Co., Ltd., Tokyo, Japan) were added to the culture. Cell proliferation was determined using the CCK-8 assay according to the manufacturer’s instructions (Dojindo, Kumamoto, Japan). After each incubation, 10 μL of CCK-8 solution were added to each well. Plates were incubated for 3 h for 37 °C, and the absorbance was read at 450 nm with a reference wavelength of 630 nm using a Varioscan Flash 2.4 microplate reader (Thermo Fisher Scientific, Waltham, MA, USA).

### 4.4. BrdU (Bromodeoxyuridine) Cell Proliferation Assay

GL261 cells were seeded onto 96-well plates at a density of 2 × 10^3^ cells/well with DMEM supplemented with 10% FBS and then incubated for 24 h, after which the culture medium was changed to DMEM containing 10% FBS. Then, astaxanthin, adonixanthin, or temozolomide (Tokyo Chemical Industry Co., Ltd., Tokyo, Japan) were added to the culture. After 72 h of culture, the culture medium was changed to DMEM containing 10% FBS and treated with BrdU at 10 µM for 3 h. After that, immunocytochemistry was performed according to the protocol of anti-BrdU antibody (abcam, ab6326).

### 4.5. Cell Migration Assay

The cell migration assay was conducted as previously described [44].

GL261 cells and U251 cells (2.0 × 10^4^ cells per well) were plated in a 12-well plate (BD Biosciences, Tokyo, Japan) with culture medium supplemented with 10% FBS. After 24 h of incubation, the medium was changed to DMEM containing 1% FBS. After 6 h, wounds were scratched by a P1000 pipette tip and washed with phosphate-buffered saline (PBS) to eliminate cell debris. Then, fresh medium was added with 10 µM astaxanthin or adonixanthin. Pictures were taken at 48 h and these scratched areas were measured using an All-in-One Fluorescence Microscope (BZ-X710; Keyence, Osaka, Japan). The cell migration rate was measured the area of the wound before migration (S0) and after migration (S1) and calculated S1/(S0−S1) × 100. The control group was then standardized to be 100%. Wound widths at 0 h were created within 1 to 1.5 mm and data were excluded if they were not suitable for that width.

### 4.6. Immunoblotting

GL261 cell was seeded at 2.5 × 10^4^ cells per well in 24-well plates with culture medium supplemented with 10% FBS. After 24 h of incubation, the medium was changed to DMEM containing 10% FBS and 0.1%DMSO PBS, 300 μM temozolomide, 10 µM astaxanthin, or 10 µM adonixanthin was added for 6 and 48 h. Cells were lysed in a special buffer (RIPA buffer R0278; Sigma-Aldrich, St. Louis, MO, USA) with a protease inhibitor cocktail (Sigma-Aldrich), phosphatase inhibitor cocktails 2 and 3 (Sigma-Aldrich), and sample buffer (Wako, Osaka, Japan). The protein concentration was determined by comparison with a known concentration of bovine serum albumin using the BCA Protein Assay Kit (Thermo Fisher Scientific). The amount of total protein was 2 µg. Equal amounts of protein in sample buffer containing 10% 2-mercaptoethanol were subjected to sodium dodecyl sulphate polyacrylamide gel electrophoresis (SDS-PAGE) in 5–20% gradient gels (SuperSep Ace; Wako), and the separated proteins were transferred to polyvinylidene difluoride membrane (Immobilon-P; Merck Millipore Corporation, Bedford, MA, USA). After blocking for 30 min with Blocking One-P (Nacalai Tesque), we incubated the membranes with primary antibodies overnight at 4 °C. The primary antibodies were a rabbit anti-phospho-p44/42 MAPK; ERK1/2 (T202/Y204) 197G2 (4377S, Cell signalling, diluted 1:1000), a rabbit anti-p44/42 MAPK; ERK1/2 (9102S, Cell signalling, diluted 1:1000), a rabbit anti-phospho-Akt (S473) 193H12 (4058S, Cell signalling, diluted 1:1000), a rabbit anti-Akt (9272S, Cell signalling, diluted 1:1000), a rabbit anti-phospho-p38 MAPK (T180/Y182) (9211S, Cell signalling, diluted 1:1000), a rabbit anti-p38 MAPK (9212S, Cell signalling, diluted 1:1000), a rabbit anti-mmp-2 (AB19167, Chemicon^®^, diluted 1:1000), a rabbit anti-mmp-9 (AB19016, Chemicon^®^, diluted 1:1000), a rabbit anti-fibronectin (ab2413, abcam, diluted 1:1000), a rabbit anti-cyclin D1 (2978, Cell signalling, diluted 1:1000), a mouse anti-p27 (sc-1641, Santa Cluz, diluted 1:500), a rabbit anti-Nox4 (NB110-5849SS, Novus, diluted 1:500), a mouse anti-Bcl-2 (sc-7382, Santa Cluz, diluted 1:500), and a mouse anti-β-actin antibody (#A2228, Sigma-Aldrich, diluted 1:1000).

After that, the membrane was incubated with the following secondary antibodies: a goat anti-rabbit IgG, or a goat anti-mouse IgG antibody (Thermo Fisher Scientific, diluted 1:1000). The band intensity was measured using an Immuno Star LD (Wako). Band intensity was measured using an LAS-4000 UV mini Luminescent Image Analyzer (Fujifilm, Tokyo Japan) and Multi Gauge Version 3.0 (Fujifilm). The phosphorylation of ERK1/2, Akt, and p38 MAPK was measured by normalizing against total ERK1/2, total Akt, and total p38 MAPK. Equal loading was confirmed using β-actin as controls for phosphoprotein signals.

### 4.7. Quantitative Real-Time Reverse Transcription Polymerase Chain Reaction Analysis (qRT-PCR)

To evaluate the effect of astaxanthin and adonixanthin on the expression of *Fibronectin* mRNA expression, we performed quantitative real-time reverse transcription polymerase chain reaction (qRT-PCR) analysis. GL261 cell was seeded at 2.5 × 10^4^ cells well in 24-well plates with culture medium supplemented with 10% FBS. After 24 h of incubation, the medium was changed to DMEM containing 10% FBS and 0.1% DMSO PBS, 300 μM temozolomide, 10 µM astaxanthin, or 10 µM adonixanthin was added for 48 h. After 48 h of treatment, RNA was isolated from GL261 cells using Nucleo Spin RNA II (Takara, Shiga, Japan). RNA concentrations were determined using NanoVue Plus (GE Healthcare Japan, Tokyo, Japan). Single-strand cDNAs were synthesized from the isolated RNAs via reverse transcription with a PrimeScript RT Reagent Kit (Perfect Real Time; Takara). Quantitative real-time RT-PCR was performed using TB Green Premix Ex Taq II (Tli RNaseH Plus; Takara) and a TP800 Thermal Cycler Dice Real Time System (Takara). All procedures were carried out in accordance with the manufacturer’s instructions. The PCR primer sequences for *Fibronectin* were as follows: 5′-CGA GGT GAC AGA GAC CAC AA-3′ (forward) and 5′-CTG GAG TCA AGC CAG ACA CA -3′ (reverse). β-actin (internal control) was as follows: 5′-CAT CCG TAA AGA CCT CTA TGC CAA C-3′ (forward) and 5′-ATG GAG CCA CCG ATC CAC A-3′ (reverse). The cycling conditions were in accordance with the manufacturer’s protocol. The results are expressed as relative gene expression levels normalized to that of β-actin.

### 4.8. Cell Death Assay

GL261 cells were seeded onto 96-well plates at a density of 2 × 10^3^ cells/well with DMEM supplemented with 10% FBS and then incubated for 24 h, after which the culture medium was changed to DMEM containing 10% FBS. Then, astaxanthin, adonixanthin, or temozolomide (Tokyo Chemical Industry Co., Ltd., Tokyo, Japan) were added to the culture. Cell death was measured by Hoechst 33,342 (Invitrogen, Carlsbad, CA, USA) and propidium iodide (Invitrogen). At 96 h after treatment, the Hoechst 33,342 and propidium iodide were added to the medium to final concentrations of 8.1 and 1.5 µM, respectively, for 15 min. Images of stained cells were captured with a Lionheart™ FX Automated Microscope (BioTek, Tokyo, Japan). The percentage of propidium iodide-positive cells was determined by distinguishing Hoechst 33,342 and propidium iodide fluorescence.

### 4.9. Reactive Oxygen Species Assay

Intracellular radical activation within GL261 cells was measured with 5-(and-6)-chloromethyl—2′,7′-dichlorodihydrofluorescein diacetate, acetyl ester (CM-H_2_DCFDA; Thermo Fisher Scientific, MA, USA). Six hours after treatment of temozolomide, astaxanthin, or adonixanthin, CM-H_2_DCFDA was added to the culture medium and incubated at 37 °C for 1 h under shading in GL261. Fluorescence was measured using a Varioscan Flash 2.4 microplate reader (Thermo Fisher Scientific, MA, USA) at 485 (excitation)-535 nm (emission). Measurements were performed 0, 30, and 60 min after the addition of CM-H_2_DCFDA.

### 4.10. Animals

All experimental design and procedures were approved by the murine experiment committees of Gifu Pharmacological University and were in compliance with ARRIVE (Animal Research: Reporting in Vivo Experiments) guidelines. These experiments were approved by the animal experiment committees of Gifu Pharmaceutical University, Japan (Ethic nos. 2018-099, 2019-065). For all experiments, male C57BL/6J mice (8 weeks old; body weight 22~27 g) and male ICR mice (6 weeks old; body weight 25~28 g) purchased from Japan SLC, Inc. (Hamamatsu, Shizuoka, Japan) were used. Animals were housed at 24 ± 2 °C under a 12-h light-dark cycle. Food and water were available to all animals ad libitum. All experimental procedures and outcome assessments were performed in a blinded manner.

### 4.11. Murine orthotopic Glioblastoma Model

Murine glioblastoma cell (GL261) transplantation was performed as previously described [39]. Briefly, mice received an intracranial injection of 1 × 10^5^ cells in 2 µL of PBS using a Hamilton microliter syringe at the following coordinates: 1 mm anterior, 2 mm lateral (left of middle) to bregma, at a depth of 3 mm from the dural surface. This protocol was completed using a stereotactic frame.

### 4.12. In Vivo Drug Treatment

In the experiment of the murine orthotopic glioblastoma model, oral administration of each astaxanthin (10 and 30 mg/kg) and adonixanthin (10 and 30 mg/kg) was initiated 3 days after intracranial injection of GL261 cells and was continued for 10 days. In the experiment of the brain tissue absorption with oral astaxanthin or adonixanthin, after one week of adaptation, ICR mice were randomly divided into the following three groups: control group, astaxanthin group (50 mg/kg), and adonixanthin group (50 mg/kg). Mice were orally administered each reagent suspended in olive oil (5 mL/kg) by the daily single dose for 10 days. The control group was treated by olive oil alone (5 mL/kg).

Since 50 mg/kg is known to be a dose of astaxanthin that does not show adverse effects even with long-term administration, this dose of astaxanthin and adonixanthin was used in the distribution experiment of this study. Moreover, it was reported that the oral administration of astaxanthin at 25 mg/kg inhibited hippocampal inflammation in diabetic mice [45]. Thus, we set up a dose similar to that in the present study.

### 4.13. Mouse Brain Analysis on In Vivo Glioblastoma Model

Mice were euthanized and transcranial perfused with cold saline for 2 min at room temperature. After that, the perfusate was changed to 0.1 M phosphate buffer (PB; pH 7.4) containing 4% paraformaldehyde (PFA, Wako Pure Chemicals, Osaka, Japan) for 3 min. Brains were fixed in 4% paraformaldehyde, embedded in paraffin (Leica Biosystems, Wetzlar, Germany), cut into 5-µm sections, and processed for haematoxylin-eosin (HE) staining. Pictures were taken using an All-in-One Fluorescence Microscope (BZ-X710; Keyence, Osaka, Japan). We assessed the maximum cross-sectional area of the tumour and tumour volume as described previously [46].

### 4.14. Collecting Blood and Tissues

In mice, blood samples were collected under anaesthesia by using sodium pentobarbital (50 mg/kg, 10 mL/kg, *i.p*). To separate serum and from blood, we centrifuged at 1700 g for 10 min. After mice were euthanized by exsanguination under deep anaesthesia, tissues, such as brain, were picked. Furthermore, the cerebral cortex, cerebellum, striatum, and hippocampus were separated from the whole brain.

All samples, including serum and tissues, were stored at −80 °C until the analysis of astaxanthin and adonixanthin was performed.

### 4.15. Analysis of Astaxanthin and Adonixanthin

Carotenoid fraction was collected from blood and some tissues by silica gel HPLC using a Cosmosil 5SL-II column with acetone:hexane (2:8, *v/v*) for the mobile phase at a flow rate of 1.0 mL/min as described above. This fraction was evaporated to dryness, dissolved in isopropanol: hexane (4:96, *v/v*), and subjected to chiral HPLC. Identification of each carotenoid was performed by authentic carotenoids obtained from Paracoccus (ENEOS Corporation, Tokyo, Japan) by our routine methods [47] and the content of each carotenoid was calculated from peak areas by comparison with the authentic samples.

### 4.16. Statistical Analysis

All data are presented as mean ± standard error of the mean (SEM). We performed the experiments assuming normality and selected an appropriate statistical analysis method depending on the presence or absence of equal variance. Specifically, student’s t-test or Welch’s test was used in the case of equal or non-equal variance under Bonferroni correction in Figure 2D, Figure 3, Figure 4, Figure 5B, and Appendix A. We used a one-way analyses of variance (ANOVA) followed by Tukey’s test or Games-Howell’s test for multiple comparisons in Figure 2A, Figure 2B, Figure 5A, Figure 6, Appendix A; and the Mann–Whitney *U*-test for two-group comparisons in Table 2 and Appendix A. These statistics were performed by SPSS Statistics (IBM, Armonk, NY, USA) software. *p* < 0.05 was considered statistical significance.

## Figures and Tables

**Figure 1 marinedrugs-18-00474-f001:**
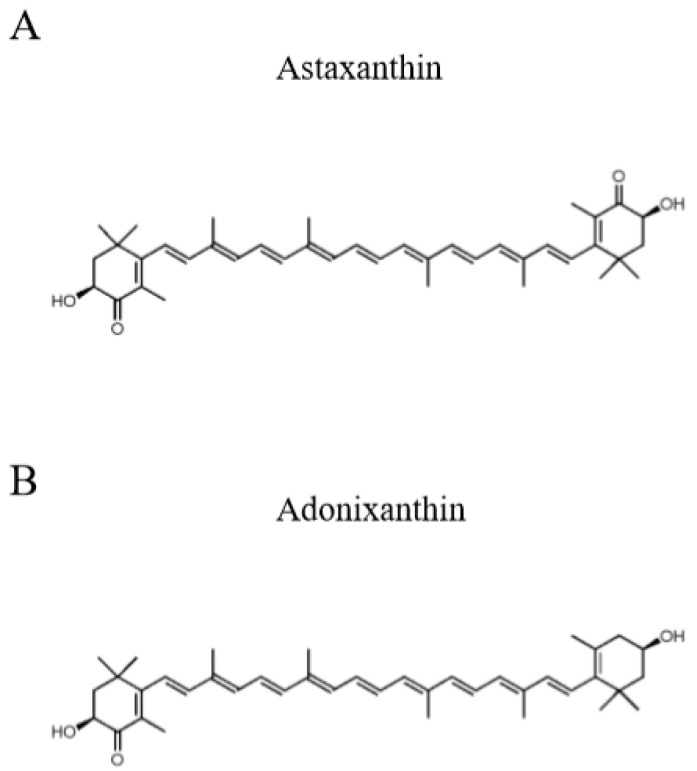
The chemical structure of astaxanthin and adonixanthin. (**A**) Astaxanthin (**B**) Adonixanthin.

**Figure 2 marinedrugs-18-00474-f002:**
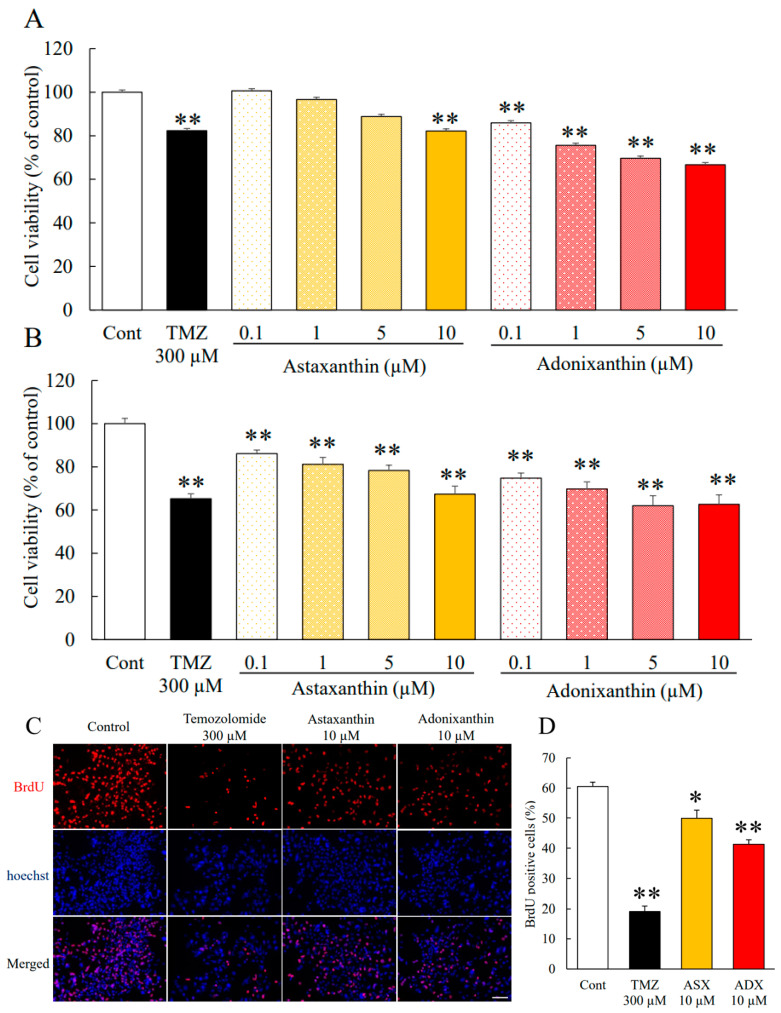
Cell viability of the mouse and human glioblastoma cell line with astaxanthin and adonixanthin. (**A**, **B**) These graphs show the cell viability of GL261 (mouse glioblastoma cell line) and U251MG (human glioblastoma cell line) treated for 96 h with temozolomide, astaxanthin, or adonixanthin. Data are shown as mean ± SEM (*n* = 6). ** *p* < 0.01 vs. control group (Tukey’s test). (**C**) These images show the representative photographs of the BrdU assay 96 h after treatment of temozolomide, astaxanthin, or adonixanthin in GL261. (**D**) This graph shows the BrdU-positive cells (%). Data are shown as mean ± SEM (*n* = 6). * *p* < 0.05, ** *p* < 0.01 vs. control group (Student’s *t*-test). TMZ; temozolomide.

**Figure 3 marinedrugs-18-00474-f003:**
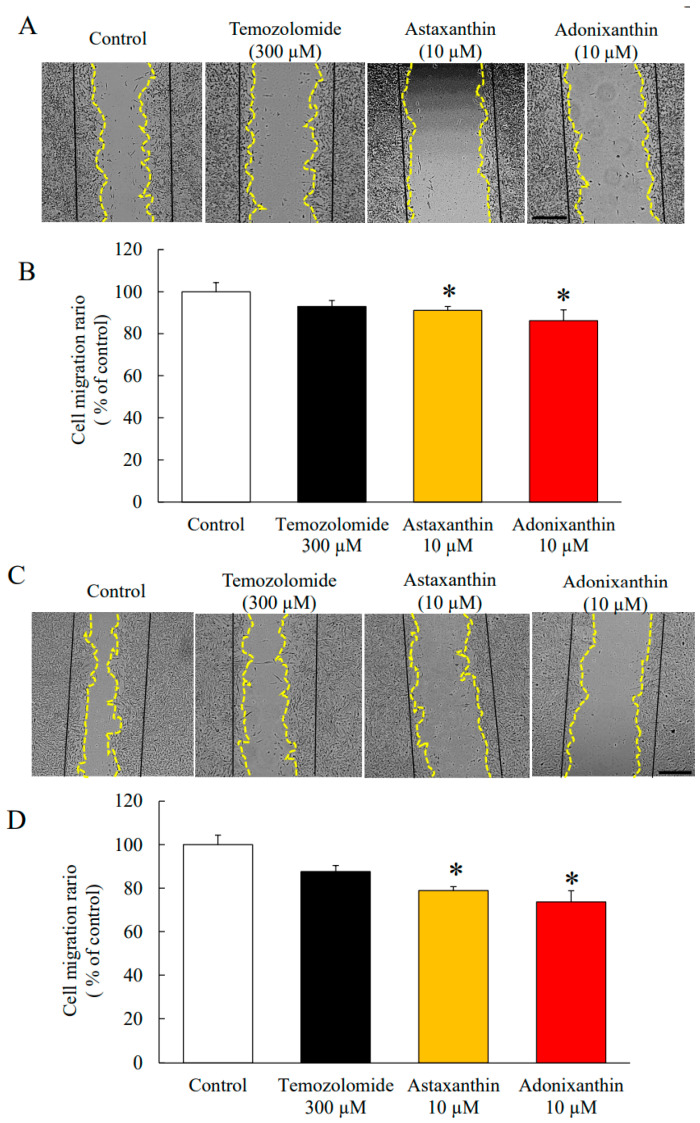
Wound healing assay with astaxanthin and adonixanthin in human glioblastoma cell U251MG and mouse glioblastoma cell line GL261. (**A**,**C**) These images show representative photographs of the wound healing assay 48 h after treatment of temozolomide, astaxanthin, or adonixanthin in GL261 (**A**) and U251MG (B). (**B**,**D**) These bar graphs show the cell migration abilities of GL261 (**B**) and U251MG (**D**). Data are shown as mean ± SEM (*n* = 3). * *p* < 0.05 vs. control group (Student’s *t*-test). Migration area ratio = (0 h scratch area—48 h non migration area)/0 h scratch area. (Original magnification × 20). The scale bars are 500 μm.

**Figure 4 marinedrugs-18-00474-f004:**
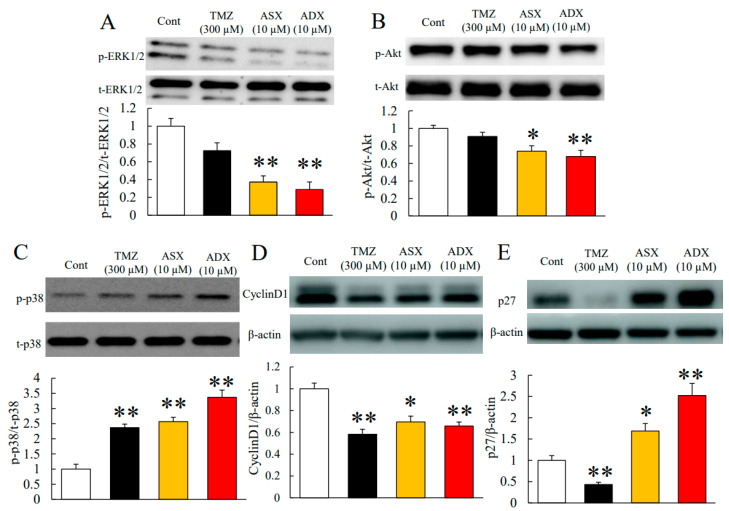
Expression of some proteins related to tumour progression after treatment of astaxanthin and adonixanthin in the mouse glioblastoma cell line. (**A**,**B**) The quantitative data of the expression levels of p-ERK1/2 and p-Akt in the mouse glioblastoma cell line GL261 at 6 h after treatment of 300 µM temozolomide, 10 µM astaxanthin, or 10 µM adonixanthin. Both astaxanthin and adonixanthin reduced the expression of p-ERK1/2 and p-Akt. Data are shown as mean ± SEM (*n* = 5 or 6). * *p* < 0.05, ** *p* < 0.01 vs. control group (Student’s *t*-test). (**C**~**H**) The quantitative data of expression levels of p-p38, p38, cyclin D1, p27, MMP-2, -9, and fibronectin, in the mouse glioblastoma cell line GL261 at 48 h after treatment of 300 µM temozolomide, 10 µM astaxanthin, or 10 µM adonixanthin. Data are shown as mean ± SEM (*n* = 6). * *p* < 0.05, ** *p* < 0.01 vs. control group (Student’s *t*-test). ASX; astaxanthin, ADX; adonixanthin, TMZ; temozolomide.

**Figure 5 marinedrugs-18-00474-f005:**
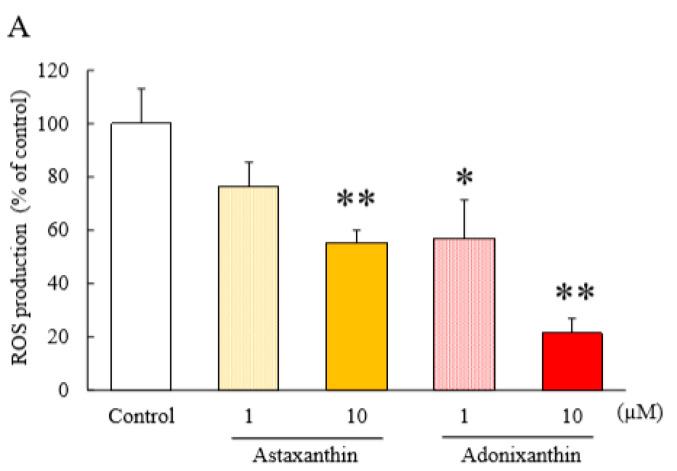
The amount of ROS and expression of Nox-4 after treatment of astaxanthin and adonixanthin in the mouse glioblastoma cell line. (**A**) ROS production was measured by an ROS detecting probe (5-(and-6)-chloromethyl—2′,7′-dichlorodihydrofluorescein diacetate, acetyl ester: CM-H_2_DCFDA). Treatment of astaxanthin and adonixanthin at 1 and 10 µM reduced the amount of ROS in GL261. Data are shown as mean ± SEM (*n* = 6). * *p* < 0.05, ** *p* < 0.01 vs. control group (Tukey’s test). (**B**) Immunoblot analysis and quantification of Nox4 at 6h after treatment of temozolomide. Data are shown as mean ± SEM (*n* = 6). ** *p* < 0.01 vs. control group (Student’s *t*-test) and ## *p* < 0.01 vs. astaxanthin 10 µM group (Student’s *t*-test).

**Figure 6 marinedrugs-18-00474-f006:**
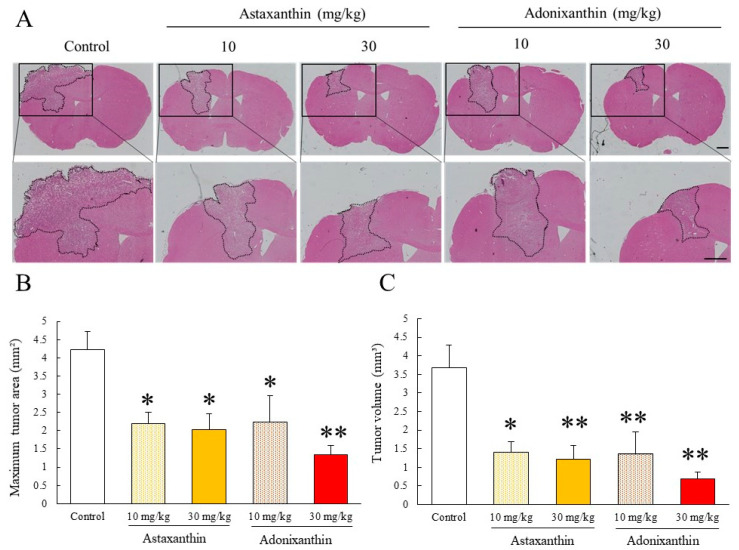
The antitumour effect of astaxanthin and adonixanthin in the murine orthotopical glioblastoma model. (**A**) These images were representative photographs of haematoxylin and eosin staining of the coronal section. Control, *n* = 11; astaxanthin 10, 30 mg/kg treated: *n* = 9, 8; adonixanthin 10, 30 mg/kg treated: *n* = 8, 10. The scale bars are 1 mm. (**B**,**C**) These bar graphs show the tumour area and volume at 2 weeks after GL261 cell injection. The oral administration of astaxanthin and adonixanthin suppressed the glioblastoma progression in an in vivo glioblastoma model. Data are shown as mean ± SEM (*n* = 8–11). * *p* < 0.05, ** *p* < 0.01 vs. control group (Tukey’s test).

**Table 1 marinedrugs-18-00474-t001:** The concentrations of astaxanthin and adonixanthin in the murine serum.

	Astaxanthin		Adonixanthin	
Compounds	Pre-treatment	4 h After the Final Oral Administration	Pre-treatment	4 h after the Final Oral Administration
*trans*-Astaxanthin	N.D.	6.04 ± 1.16		
*cis*-Astaxanthin	N.D.	13.4 ± 3.78		
*trans*-Adonixanthin			N.D.	34.7 ± 3.00
*cis*-Adonixanthin			N.D.	35.02 ± 5.01

Data are shown as mean ± SEM (*n* = 4). ng/mL. N.D.; not detected.

**Table 2 marinedrugs-18-00474-t002:** The concentrations of astaxanthin and adonixanthin in the murine tissues.

Brain Tissues	Astaxanthin		Adonixanthin	
	*trans*	*cis*	*trans*	*cis*
Cerebral cortex	5.22 ± 0.87	3.62 ± 1.56	3.24 ± 0.44 *	N.D.
Cerebellum	5.06 ± 1.85	2.43 ± 1.58	2.85 ± 1.05	N.D.
Striatum	11.37 ± 3.14	3.76 ± 2.06	2.90 ± 1.87	N.D.
Hippocampus	12.28 ± 1.07 *	2.61 ± 1.33	3.63 ± 2.10	N.D.

Data are shown as mean ± SEM (*n* = 4). ng/g. * *p* < 0.05 vs. *cis*-forms group (Mann–Whitney *U*-test). N.D.; not detected.

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
