# Peer review of "Antitumour Effects of Astaxanthin and Adonixanthin on Glioblastoma"

_marinedrugs, 2020, doi:10.3390/md18090474_

Round 1

Reviewer 1 Report

This is a resubmission of the manuscript which has been previously rejected from Marine drugs. The authors have performed the requestedexperiments and have added the required information. The manuscript is interesting and contains both in vitro and in vivo data on the antitumor activity of astaxanthin and adonixanthin in GBM models.

Even though the manuscript was significantly improved, I still miss the detailed protocol of the detection of astaxanthin and adonixanthin in the brain and blood. Was MS analysis used as well? If not, then how did the authors identified a peak corresponding to one of the investigated compounds? Were the standards used?

In the introduction, a more detailed description of the mechanism of action is required; especially the targets that may be related to the anticancer activity of astaxanthin and adonixanthin. Additional to your own articles, please also cite the other sources. 

Reviewer 2 Report

In the manuscript, Tsuji and colleagues assayed the effects of astaxanthin and adonixanthin on glioblastoma in vitro and in vivo models. Glioblastoma (GBM), which forms from astrocytes, is the most dangerous and aggressive form of brain cancer. GBM patients typically have short life expectancies after diagnosis. Only a quarter of newly diagnosed GBM patients survive for 24 months, and fewer than 10 percent of patients survive more than 5 years. In the manuscript, the Authors focused on a xanthophyll carotenoid, astaxanthin, and its intermediate product, adonixanthin. Both drugs can cross the BBB and  have anti-tumour properties. In particular, in this study, astaxanthin and adonixanthin inhibited both cell proliferation and migration in human and mouse glioblastoma cell and exhibited an anti-tumour effect in glioblastoma mouse model.

Given the importance of finding new potential treatments for glioblastoma, this paper could be interesting for the scientific community and overall, the findings are interesting. On the other hand, unfortunately, this manuscript suffers several major concerns that should be addressed before further consideration.

  1. The Authors indicate that the anti-tumour effect of both astaxanthin and adonixanthin may be mediated by not cell death, but cell cycle arrest, because the treatments decreased the level of cyclin D1 but not BCL2. It is necessary to confirm this result with other assays, such as cell cycle analyses and/or other protein involved in cell cycle.
  2. The Authors performed several treatment time points for the different assays. i.e. 96h for the Viability assay, 48h for the migration, 6h for western blotting analysis and 72h for death assay. The Authors should standardize the treatment time, in particular, it would be interesting to have the viability results at different time points (6h, 24h, 48h and 72h) and/or they should justify the choice.

Furthermore, there are minor issues:

  1. In the figures 1,2,3 and 4 the order of the treatments in the graphics are different, in particular for the temozolomide.
  2. In the in vivo study, it would be better if the temozolomide group was present as a positive control
  3. In line 269 is indicated “cell proliferation”, but it is a viability assay.

Round 2

Reviewer 2 Report

I would like to thank the Authors for addressing the critical points. The manuscript is now suitable for publication.

Author Response

I have already answered to Reviewer 2's comments.

This manuscript is a resubmission of an earlier submission. The following is a list of the peer review reports and author responses from that submission.

Round 1

Reviewer 1 Report

The manuscript by Tsuji et al. presents an interesting data on antitumor activity of astaxanthin and adonixanthin on glioblastoma model. Although the data are promising, the manuscript itself should be improved before it can be accepted for publication.

Major points:

  • The authors should examine the effect of ASX and ADX treatment on the apoptotic / cell death markers (caspase-3 activation; PARP; phosphatidylserine externalization), as the observed downregulation of p-ERK and p-Akt, as well as MMP-2 might be an unspecific effect of the cell death related processes. The results should be discussed in the Discussion chapter.
  • I would also suggest to check the effect on the cell cycle progression in order to access anti-proliferative activity of the compounds
  • The language stile of the manuscript should be improved

Minor points:

  • From my point of view, the Discussion is too superficial and lacks the analysis of the data generated in the current research.
  • The values (e.g. page 3, line 82, as well as overall the manuscript) should be presented as mean +/- SD.
  • The ROS down-regulation may be related to the anti-oxidant activity of the compounds. This should be discussed in the Discussion chapter.
  • In the chapter 2.6 please represent the mice weight as a graph.
  • Please explain how the doses for in vivo experiments were chosen.
  • Fig 6a – it is not clear for the Figure legend which doses we used for each group.

Author Response

Thank you for your suggestion.

Reviewer 2 Report

In this manuscript, Tsuji et al. investigated the anti-tumour mechanisms of a xanthophyll carotenoid, astaxanthin, and its intermediate product, adonixanthin, as potential treatment for glioblastoma. The study design is appropiate and the results are of novelty.

However, the study present a great major concern: All the in vitro experiments have been performed only in one murine cell line (GL261) and one human cell line (U251MG). Given the heterogenicity present in glioblastoma tumors, it would be necessary the inclusion of new glioblastoma cell lines to increase the robustness of the in vitro results.

It is also recommended to perform qPCR to corroborate the results obtained by immunoblotting in the expression of some proteins that promote cell growth and migration.

I encourage the realization of the proposed new experiments and the resubmission of the manuscript.

Author Response

Thank you for suggestions.

Round 2

Reviewer 1 Report

I thank the authors for the informative rebuttal letter. However, I have to admit that the authors filed to answer most of the raised questions. For example, the induction of apoptosis and alteration of the cell cycle related proteins was shown by the other groups in different cancer model (hepatocarcinoma, leukemia, colon cancer etc.). Thus, a) it does not mean at all that the same effects will take place in GBM model used in this research and b) it does not mean that the active concentrations which will cause the same effects (if any) in GBM model are comparable to the one authors used for example for the ERK1/2 inhibition / MMP-2 downregulation in the current research. Additionally, the activation/inhibition of ERK1/2 and Akt may affect numerous processes, and can be either anti- or pro-apoptotic (i.e. anti- or pro-cancer), depending on the model used and the stimulus nature. Therefore based on the current data the authors can not postulate that the inhibition of ERK1/2 and Akt phosphorylation is sufficient to induce apoptosis.

Therefore I unfortunately can not recommend this manuscript for publication in the current form. At the same time I appreciate the value of the in vivo data presented in the current manuscript.

Please add the explanation of the dose selection in the manuscript body. Please cite the related references if the decision was done based on the previously published data.

For the Fig. 6 legend I would recommend the following way of the doses representation: “Control, n = 11; astaxanthin 10, 30 mg/kg treated: n = 9, 8; adonixanthin 10, 30 mg/kg treated: n = 8, 10."

Additionally:

Why Fig. 1 is plates in the manuscript after the Fig. 6?

I still believe that the language style should be improved by the native speaker familiar with the field. Just as an example, “Furthermore, these compounds were confirmed to migrate in a murine model” means that the compounds are able to cross the BBB, or to distribute from blood to the tissues, or to “migrate” from the stomach to the blood?

Reviewer 2 Report

Thanks to the authors for the response. However, prior concerns still present. Realization of proposed new experiments are mandatory to improve the scientific soundness of the results.